# Revealing the Differences in *Ulnaria acus* and *Fragilaria radians* Distribution in Lake Baikal via Analysis of Existing Metabarcoding Data

Alexey Morozov *, Yuri Galachyants, Artem Marchenkov, Yulia Zakharova and Darya Petrova

Limnological Institute, Siberian Branch of the Russian Academy of Sciences, 3 Ulan-Batorskaya St., Irkutsk 664033, Russia
* Correspondence: morozov@lin.irk.ru; Tel.: +7-902-1765206

**Abstract:** Two diatom species, *Ulnaria acus* and *Fragilaria radians*, are morphologically very similar and often coexist, which makes it difficult to compare their abundances. However, they are easily separated by molecular data; thus, in this work, we attempted to estimate the differences in their spatial and temporal distribution from existing metabarcoding datasets. Reanalyzing published sequences with an ASV-based pipeline and ad hoc classification routine allowed us to estimate the relative abundances of the two species, increasing the precision compared to usual OTU-based analyses. Existing data permit qualitative comparisons between two species that cannot be differentiated by other methods, detecting the distinct seasonal peaks and spatial distributions of *F. radians* and *U. acus*.

**Keywords:** metabarcoding; hidden diversity; diatoms; phytoplankton





## 1. Introduction

In Lake Baikal, like in other freshwater ecosystems, diatoms play a significant role in primary production, sediment deposition and biogeochemical cycles. Among the phytoplanktonic species of the lake, one of the dominant species was originally identified as *Synedra acus* subsp. *radians* Skabitsch. and later renamed *Fragilaria radians* (Kützing) D.M.Williams and Round [1]. This diatom (referred to as *F. radians* in the Introduction, regardless of the name used in other papers) is not only a major player in the lake ecosystem, but also a model object for multiple studies. In particular, it was successfully axenized [2], which allowed it to become the first freshwater diatom to have its nuclear genome sequenced [3].

Since *F. radians* is a key element in the Lake Baikal ecosystem, its dynamics and ecology have been thoroughly studied. This alga blooms under ice, both in littoral and pelagic areas, dominating the eukaryotic phytoplankton community associated with the lower ice surface [4,5]. After the ice-breaking period (April to May), the *F. radians* population decreases; however, it still remains a significant member of the phytoplankton community until mid-summer [6–8]. Many correlations were found between *F. radians* abundance (as estimated by either microscopy or metabarcoding studies) and various biotic and abiotic factors at different times and sampling stations [8–10]. Usually, its abundance and biomass correlate negatively with Si availability and positively with abundances of other common Baikalian diatoms, although there are exceptions [11].

In a recent study, this population, previously thought to consist of a single *Fragilaria* species, was found to include members of two species from different genera: *Fragilaria radians sensu stricto* and *Ulnaria acus* (Kützing) Aboal. The morphology of these two species is nearly identical: *F. radians* has a cell length of 105–239 µm, cell width of 2.5–5.2 and 12–22 rows of areolae per 10 µm; U. acus has a cell length of 60–251 µm, cell width 2.2–5.4 µm and 12–14 rows of areolae per 10 µm. They can also be cultured under identical

conditions and have been isolated together from natural samples. This makes separating them in routine microscopy-based phytoplankton monitoring next to impossible, since it requires either sophisticated EM-based analyses or DNA sequencing [12]. All of the ecological studies listed above were also based on the assumption of a single species, using either light microscopy or wider *Synedra* sp./*Fragilaria* sp. OTUs that may have included a mixture of reads from *U. acus*, *F. radians* or other related species. Further, the taxonomy of these OTUs was typically not identified below the genus.

These similarities explain why the two species have not been separated until recently. However, similar morphology and overlapping ranges of acceptable conditions do not imply exactly identical autecological features of the two species. It is possible that, although overlapping, the optimal temperatures or other factors are somewhat different for the two species. On the other hand, freshwater benthic *Fragilaria* and *Ulnaria* strains identified from molecular data in various streams and lakes in Europe showed considerable overlap in geographical distribution, habitat and ecological preferences [13]. Although multiple OTUs of Fragilaria sp. and Ulnaria sp. were observed in most studies on Lake Baikal, the issues outlined above render them unsuitable for discussing the ecology of these two species.

Thus, a goal of this work was to develop a method to separately estimate the relative abundances of *U. acus* and *F. radians* based on metabarcoding data, and to apply this method to the existing sequences from Lake Baikal.

## 2. Materials and Methods

### 2.1. Selecting Amplicons for Distinguishing F. radians and U. acus

In order to study whether 18S rRNA or rbcL metabarcoding analysis can distinguish all groups within *Ulnaria*/*Fragilaria* species complex, metabarcoding libraries were produced from two samples of phytoplankton taken near the settlement of Bolshiye Koty in March 2020, as well as a mock community consisting of four strains isolated from Lake Baikal. In order to extract DNA, integral water samples of 20 L (equal volumes of samples from different depths) were collected. Samples were first pre-filtered using a 27 μm sieve and then were filtered through 0.2 μm analytical track membranes (Reatrack, Obninsk City, Russia). Biomass was washed off the filters into sterile TE buffer (10 mM Tris–HCl, 1 mM EDTA, pH 8.0) and stored at −80 °C until analysis. DNA was extracted using lysozyme (1 mg mL$^{-1}$), proteinase K, 10% SDS and phenol:chloroform:isoamyl alcohol mixture (25:24:1) according to the protocol based on Rusch et al. [14].

The mixture included strains L150 (*F. radians*), MM244 (*U. acus*), BZ264 (*U. ulna*) and ChZ411 (*Aulacoseira islandica)* from Lake Baikal. In order to isolate the *Fragilaria* and *Ulnaria* monoclones, phytoplankton samples were collected in different parts of Lake Baikal during 2017 and 2018. The latter two strains were included because *U. ulna* and *A. islandica* commonly coexist with our species of interest; therefore, any practically useful method should be able to distinguish them from *U. acus* and *F. radians*. Cultures of diatoms were obtained via isolation of individual cells. The isolated strains were grown in 96-well plates with Diatom Medium (DM) in a mini-incubator at 8 °C and illuminated with 16 μL Einstein m$^{-2}$ s$^{-1}$ at a photoperiod of 12:12 h light:darkness, and then transferred into Erlenmeyer flasks with a volume of 100 mL for further growth. The strains were grown for three months to obtain sufficient biomass for DNA extraction.

Approximately 300,000 cells were taken for each of the three *Ulnaria* and *Fragilaria* strains. The cell number for the colonial species *A. islandica* was not known precisely; however, a roughly similar number of cells was taken. DNA was isolated as described above and amplified using two primer pairs for each target gene. V3-V4 18S rRNA (418 bp) was amplified with TAReuk454FWD1 5′-CCAGCASCYGCGGTAATTCC and TAReukREV3 5′-ACTTTCGTTCTTGAT [15] primers. For V8-V9 18S rRNA (368 bp), we used V8F 5′-ATAACAGGTCTGTGATGCCCT and 1510R 5′-CCTTCYGCAGGTTCACCTAC [16]. Both 18S rRNA fragments were amplified with Phusion Hot Start II High-Fidelity DNA polymerase (Thermo Fisher Scientific, Waltham, MA, USA). PCR mix consisted of 1× Phusion buffer HF, 1 unit of DNA polymerase, 0.2 mM dNTP mixture, 1.5 mM free Mg$^{2+}$, 0.2 μM

primers and 10 ng of DNA. Temperature profile was as follows: 98 °C for 1 min, 29 cycles of (98 °C for 30 s, 50 °C for 30 s, 72 °C for 30 s) and 72 °C for 3 min. PCR product was purified by AMPure XP magnetic beads (Beckman Coulter, Brea, CA, USA) according to the manufacturer's protocol.

A 312 bp rbcL fragment amplified by pairDiat_rbcL_708F 5′-AGGTGAAGTTAAAGGT TCATACTTDAA [17] and R3 5′-CCTTCTAATTTACCAACAACTG [18] primers was previously proposed for diatom metabarcoding [19]. Since this primer pair has some mismatches with rbcL sequences produced in previous work [12], we designed an additional primer pair (bar_S_rbcL_665F 5′-GCAACAGGTGAAGTTAAAGGTTCT and bar_S_rbcL_867R 5′-GAGTTACCTGCACGGTGTAAGT) to amplify the Baikalian *Fragilaria* and *Ulnaria*. These two primer pairs are referred to as rbcL606 and rbcL708, respectively, in the remaining text. PCR with both rbcL primer pairs was performed using Tersus polymerase (Evrogen, Russia). PCR mix consisted of one Tersus Red buffer, 1 unit of DNA polymerase, 0.2 mM dNTP mixture, 0.2 µM of both primers and 100 ng of DNA. Temperature profile was as follows: 95 °C for 3 min, 30 cycles of (95 °C for 30 s, 60 °C for 30 s, 72 °C for 30 s), 72 °C for 5 min. PCR product was analyzed by electrophoresis in 1.5% agarose gel and purified with Monarch Gel Extraction Kit (NEB, USA) according to the manufacturer's protocol.

Libraries were sequenced on Illumina Miseq with MiSeq® Reagent Kit v3 (2 × 300 bp) in the Core Centrum "Genomic Technologies, Proteomics and Cell Biology" in ARRIAM (All-Russia Research Institute for Agricultural Microbiology, Russia). Thus, 18S rRNA amplicon libraries were analyzed in mothur 1.44.11 [20] to produce 97% identity OTUs and ASVs, as well as in Usearch 11.0.667 to produce ASVs using the unoise3 algorithm. In both cases, ASVs were generated at a cutoff of 4 substitutions. In mothur-based analysis, the Silva nr v138.1 database was used as a reference for alignment and taxonomic classifications. Since this database does not offer taxonomic resolution below genus, all OTUs/ASVs assigned to genera *Ulnaria* and *Fragilaria* were BLASTed against 18S rRNA sequences sequenced from Baikalian diatoms [12] using blastn 2.2.31+ [21]. Those with sequence identities exceeding 97% with all sequences from one clade, but not others, were classified as belonging to corresponding groups. OTUs and ASVs, which had high-quality hits with both *Ulnaria* clades, were classified as *Ulnaria* sp.; any other combination of hits was considered unclassified. For the purposes of this classification, *U. ulna* and *U. danica* reference sequences were treated as a single group, since these two species are hard to distinguish from either morphological or genetic data and elucidation of their relationship is beyond the scope of this paper.

The rbcL amplicon libraries were analyzed with Usearch only; *Fragilaria*/*Ulnaria* ASVs were classified in a similar way using 98% identity cutoff. All sequencing data are available at NCBI SRA (project ID PRJNA666300).

### 2.2. Analysis of Published Metabarcoding Data

Raw reads and sample metadata were downloaded from the public databases (ENA Project ID PRJEB24415 for 2013 spatial dataset [22], NCBI SRA project IDs PRJNA657482 and PRJNA662681 for 2013 spatial dataset [23] and 2017 time series [8]). Only V4 amplicons were selected from the latter dataset; otherwise, all available data were used. Usearch and vsearch [24] were used to filter reads (maximum expected error 1.0, minimum assembled length 400 bp), produce ASVs with usearch UNOISE algorithm and remove the chimerae with vsearch UCHIME. To estimate the abundances of these ASVs, filtered reads were mapped to them at 99% identity cutoff using usearch. Preliminary taxonomic annotation was produced by kmer-based naive Bayesian classifier implemented in mothur [20] with SILVA v138.1 reference alignment and taxonomy [25]. These analyses were performed separately for each dataset; the pipeline was identical to that described above for testing datasets.

All ASVs identified as *Ulnaria* or *Fragilaria* were classified as described above. Data processing and plotting were performed in Python3 using matplotlib [26] and Basemap packages.

### 3. Results

#### 3.1. Amplicon Selection and Testing

Clustering the reads from a mock community into OTUs produced a questionable result, with multiple OTUs per species. These OTUs were absent from natural samples. We assumed that these are clustering artifacts. Further, the total abundance of classified OTUs was very low in the mock sample. Because of this, as well as the conceptual arguments in favor of ASVs/zOTUs over OTUs (28), further analysis was carried out in terms of ASVs. Usearch-produced ASVs were classified with more precision than mothur-produced ones (no Usearch mock community ASVs were assigned to "*Ulnaria* sp." and "unknown"); therefore, only the results of the Usearch ASV pipeline are documented below (all ASVs and abundances are available in Supplementary Table S1).

Complete taxonomies for all 18S rRNA amplicons are available in Supplementary Table S2. In a mock community and in natural samples, V8-V9 variable regions of the 18S rRNA gene were not able to distinguish between the *Ulnaria acus* and *Ulnaria ulna/danica* clades. Proportions of studied groups in the libraries of V3-V4 18S rRNA and both rbcL amplicons are shown in Figure 1. Analysis of a mock community does not recover all three groups at equal abundances; *Aulacoseira islandica* is also strongly overrepresented in V3-V4 and V8-V9 libraries (Supplementary Table S2). The four tested marker/pipeline combinations also do not produce exactly identical results. However, all tested markers—except V8-V9 18S rRNA regions—appear to be applicable for studying the relative abundance of *U. acus* and *F. radians*, and they do not wildly contradict each other.

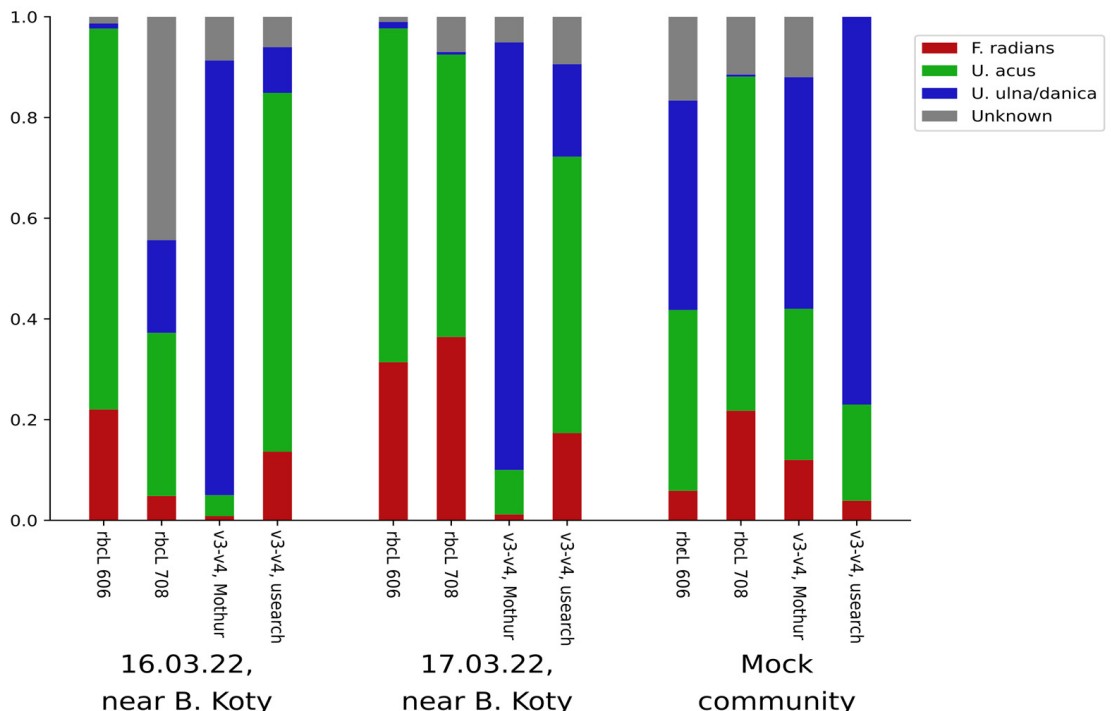

**Figure 1.** Relative abundances of *F. radians*, *U. acus* and *U. ulna/U. danica* in the sample from Lake Baikal, near Bolshiye Koty settlement, and culture mixture, as revealed by V3-V4 18S rRNA and rbcL 606 amplicons. Although the mock community contains only *U. ulna*, the classification pipeline does not distinguish it from *U. danica*; therefore, the ASVs of this species are marked as *U. ulna/U. danica*.

*3.2. Analysis of Existing V4 Datasets*

As shown above, V8-V9 variable regions of 18S rRNA are not suitable for studying Baikalian populations of *U. acus* and *F. radians*. rbcL could potentially be useful, but no sequencing data for this amplicon in Lake Baikal samples are publicly available. As for V4 18S rRNA amplicons, three relatively large datasets exist for Lake Baikal: a time series of water column samples from 0–25 m depths during March–September 2017 taken in the pelagic zone of the Southern Basin of Lake Baikal [8], and two datasets from multiple sites and depths within the lake sampled in July 2013 [22] and in the summer of 2017 [23].

Although all three datasets consist of Illumina MiSeq reads, they were amplified with three different primer pairs targeting slightly different 18S rRNA fragments. The amplicon used in [23] failed to produce ASVs that map with 99%+ identity to *Fragilaria radians*. Although there is a number of ASVs that align to *F. radians* better than they do to the two *Ulnaria* species (at roughly 97.5% identity), these sequences could potentially belong to *Fragilaria* species other than *F. radians*, which are known to be present in Lake Baikal [27,28]. Therefore, this dataset was excluded from further analysis, leaving us with one time series from 2017 and one geographical series from 2013.

As Figure 2 shows, the seasonal dynamics of both species follow the general pattern previously documented for *Fragilaria radians sensu lato* (see Introduction), with a spring bloom followed by a decrease in summer, and the near-complete absence of these diatoms in autumn. However, *U. acus* lags behind *F. radians sensu stricto* by roughly a month. Figure 3 shows that the peaks of both populations are positioned very close to the end of the ice season. There are no samples available for the melting period itself (late April to mid-May), but the highest relative abundance of *F. radians sensu stricto* predates this period, while the *U. acus* population peaks in open water.

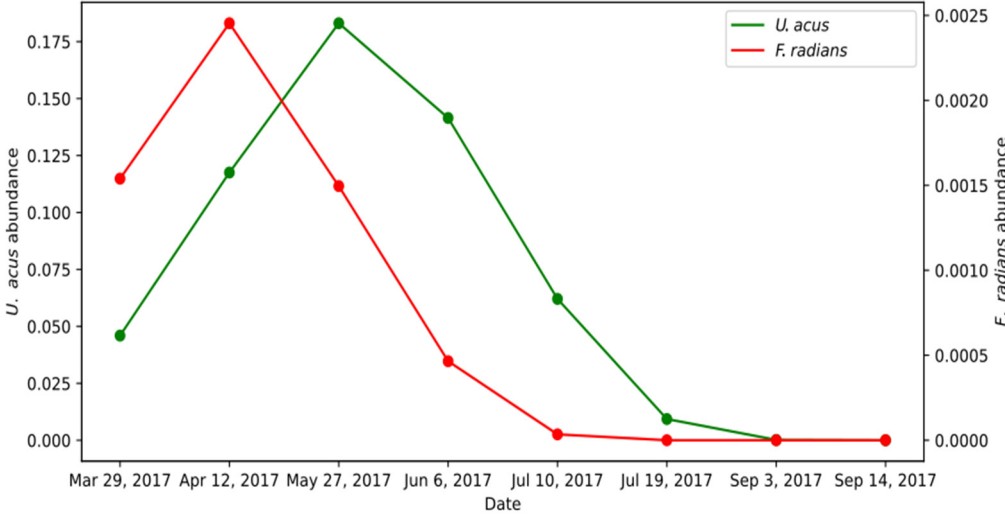

**Figure 2.** The relative abundance of *U. acus* and *F. radians* in the 2017 time series. Relative abundances of both species (the number of reads identified as one or the other, divided by library size) are plotted on separate Y axes against the time.

Distribution in the lake, as recovered from the 2013 dataset [22] (Figure 3), is also not identical. The highest abundance of both species is identified at the edge of Southern and Central Baikal, near the estuary of Selenga River. Both are also present, although in lower numbers, between Olkhon Island and Svyatoy Nos Peninsula; however, neither has been identified in the Southern Basin. The species distribution differs in the North: while *U. acus* is barely present in this area, *F. radians* populations are similar in size to most of those in Central Baikal. There is no clear pattern for their distribution along depth; however, this distribution is also not identical for most sampling sites.

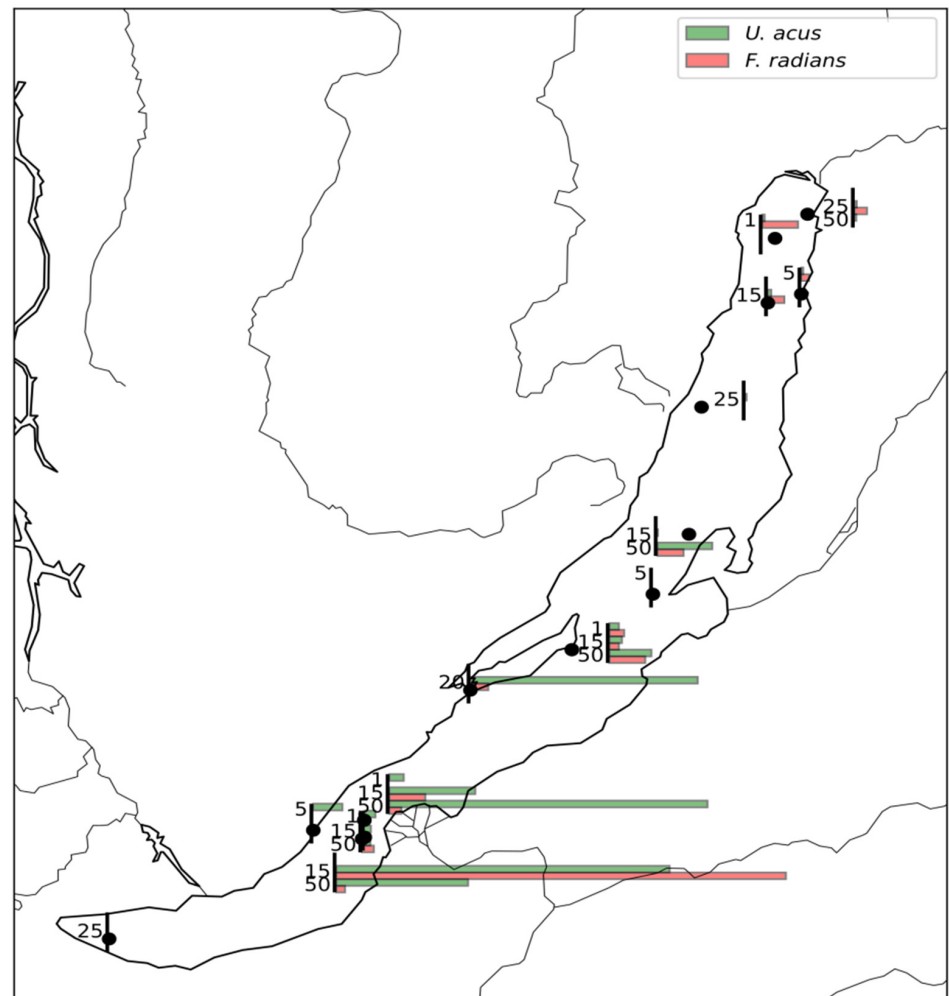

**Figure 3.** The relative abundance of *U. acus* and *F. radians* in the spatial distribution series; sampling depths are marked. Relative abundances of two species are not to scale with each other.

## 4. Discussion

### 4.1. Identifying the Target Species in Existing Data: Methodological Discussion

We estimated the relative abundances of two otherwise nearly indistinguishable species using available metabarcoding data. In addition to the result itself, the study may also be relevant because we encountered several pitfalls that may appear in similar works in the future.

First of all, a commonly used V8-V9 18S rRNA amplicon failed to separate *U. acus* from related *U. ulna* and *U. danica* species usually coexisting with it in Lake Baikal. This is unsurprising, because primer design in metabarcoding studies involves a tradeoff between taxonomic coverage (amplifying the barcode from as many taxa as possible), which requires a conservative sequence region, and precision (ability to separate closely related species), which benefits from having as many substitutions as possible. Commonly used primer pairs are designed to hit the sweet spot of amplifying the majority of eukaryotes while still being able to identify at least some genera [19]. OTU-based bioinformatics pipelines also usually target this kind of resolution, using 97% or 99% identity cutoffs that roughly correspond to species or genus but may also include several taxa (which is, in fact, where the term "Operational Taxonomic Unit" comes from—there is no guarantee that all OTUs generated at a given identity threshold correspond to taxa of a certain taxonomic rank).

This framework is useful for describing the overall composition of communities because ecological differences within genera are usually considered less important, while missing some large distant group altogether will heavily affect the conclusions. However,

the requirements of our study are exactly the opposite; we are interested only in a small group of taxa, but with as high a resolution as possible. Practically, this difference in requirements means two things. First, any given amplicon may or may not work, regardless of how useful it was for previous studies of the same ecosystem. Second, the bioinformatics pipeline needs to be optimized for precision.

The obvious first step in this optimization is to use ASVs/zOTUs rather than larger OTUs [29]. Although they do not directly correspond to the taxa as well, ASVs are intended to be as small as possible, which means that any given ASV will correspond to some genotype or strain within species, rather than a group of species. Even with ASVs, the identification procedure has to involve an ad hoc pipeline with a custom reference database, because existing wide-range taxonomic databases do not have resolution below a genus level [25].

Even if relative abundances of the taxa in question have been estimated, it is well known, both from the literature [29] and from the mock community test in this work, that read counts correlate poorly with the actual biomass or cell count of the corresponding species. Further, the different datasets (even with the same marker gene) are produced with somewhat different methods and, therefore, are poorly compatible. On the other hand, it should be noted that although these differences are likely to be smaller in magnitude than those between different taxonomists using microscopy [30], they are not guaranteed to be small enough for quantitative comparison. Although it is tempting to call for the unification of methods that would allow for seamless co-analysis of datasets, this unification may have an unexpected downside if a universally accepted amplicon is unsuitable for some narrow problem (as V8-V9 18S rRNA was in our work). Diversity of methods, on the other hand, provides the chance that at least some part of the existing data would fit the requirements of any study.

### 4.2. Autecology of U. acus and F. radians

*U. acus* and *F. radians* were shown to exhibit both temporal and spatial differences in distribution. In 2017, in Lake Baikal, they follow a similar annual trend (Figure 2), but *F. radians* passes each stage of this trend before *U. acus*. Both under-ice blooms and post-melting populations are probably formed by a mixture of both species, but it appears that *F. radians* numbers start decreasing approximately when the ice starts melting. *U. acus*, on the other hand, continues the bloom and peaks in open water. In summer, both populations continue to decline, with *U. acus* lagging behind *F. radians*.

Spatial distribution is only observed in July, which is a period of decline for both species (as can be seen both in published data [9] and from the 2017 time series). However, both this decline and the lag identified from the time series fail to explain the observed spatial distribution in July 2013. Neither species monotonously decreases along the north–south axis, as would be predicted by a simple model where the Northern Basin lags behind Central and Southern Baikal in seasonal changes. There is also a difference in their distribution along the depth of the water column, but it is not similar across stations. Further, all samples are taken within the photic layer, ignoring the sub-photic zone, which makes it difficult to discuss the possible vertical migration.

In both datasets, the ASV abundance ratio is skewed towards *U. acus* by one or two orders of magnitude (Figure 4). Although analysis of 18S amplicons from the mock community was shown to overestimate *U. acus* abundance (or, equivalently, underestimate *F. radians* abundance), this overestimate was below an order of magnitude. Analysis of natural samples from the Bolshiye Koty settlement has shown a similar bias, although without a better estimate of real abundances, this bias could not be quantified. Further biases could be introduced by the ad hoc classification procedure used in this work. If, for example, some subpopulation of either species is sufficiently divergent for its 18S sequence to be less than 99% identical to reference strains, this would also lead to underestimating the abundance of the species in question.

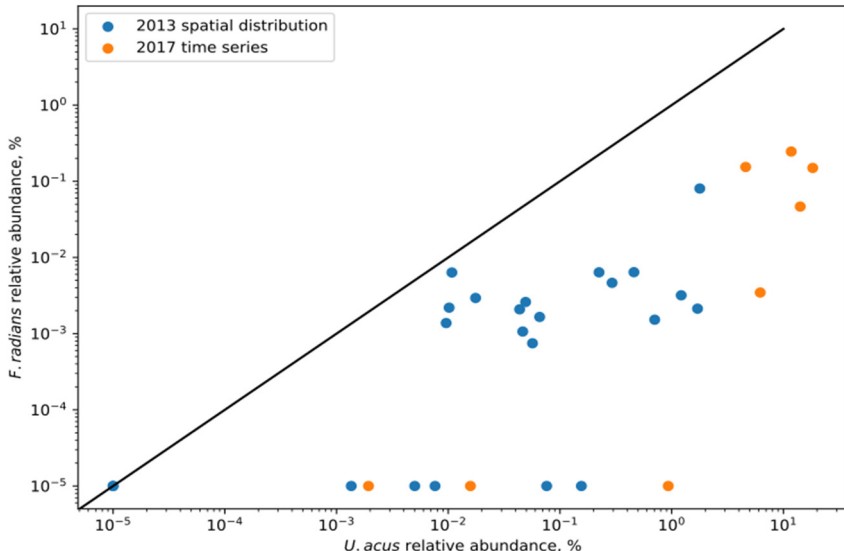

**Figure 4.** The ratio of *F. radians* and *U. acus* relative abundances in all studied samples. Both axes are in log scale; a pseudocount of $10^{-5}$% was added to all values to make zeroes visible in log axes. Black line shows the 1:1 ratio.

Another reason to believe that the difference in abundance is an overestimate of orders of magnitude comes from the fact that the number of strains of both species in the laboratory collection of Limnological Institute is roughly similar [12]. At the time that these strains were isolated, the object of study was considered a single species, so no specific measures were taken to preferentially cultivate *F. radians*. However, it is possible that *U. acus* is significantly less viable in culture, leading to an unintentional enrichment in *F. radians*.

Between the unknown biases introduced by amplicon sequencing, ASV identification and culturing, our amplicon testing data cannot be directly used to calibrate the point estimates of a *Fragilaria/Ulnaria* relative abundance ratio. Even if such a calibration was possible, its results would only apply to the tested amplicon (identical to the one in the 2017 dataset [8]), not necessarily extending to the amplicon used in the 2013 data [22]. Thus, we cannot produce quantitative estimates of the two species' abundances, or even their ratio, from metabarcoding data. We can only qualitatively conclude that *U. acus* is likely more abundant than *F. radians*, but we cannot make claims about the magnitude of this difference.

However, we can at least assume that the same species within the same analysis is subject to roughly the same artifacts in all samples. Using this assumption, it is possible to compare the distribution of the species throughout space and time. In other words, one can use the 2017 time series to see whether the seasonal dynamics of the two species were similar in 2017 at the Listvyanka–Tankhoy transect, while the 2013 spatial dataset can be used to see whether they were distributed similarly between various sampling sites and depths across Lake Baikal in July 2013.

Using this assumption, we can observe that the two species exhibit differences in distribution, which, in turn, implies autecological differences. It is tempting to suggest that *F. radians*, which blooms earlier and remains abundant for a longer time in Northern Baikal, is more psychrophilic (or at least psychrotolerant, considering that both are cultured successfully at higher temperatures).

However, any ecological conclusions made from the two relatively small datasets used in this work would be speculative at best, and they may be compromised by the methodological concerns discussed above. In addition, metabarcoding data do not distinguish active and resting cells, and there is a precedent of inactive *F. radians* cells observed in near-surface water in July 2019. These cells have probably finished their bloom and sunk below the photic layer, only to be returned via upwelling [11]. Although such events are thought to be rare, there is no guarantee that something similar did not happen in Northern Baikal in 2017.

A more detailed investigation of these species' distribution requires larger volumes of data than are currently available. Further, the existing data only focus on a small number of abiotic factors, while the difference between species may be something obscure such as: a resistance to an unevenly distributed pathogen or grazer; different light requirements; or sensitivity to minor changes in water chemistry. Similar conclusions were reached in [13], which also used existing metabarcoding datasets to study the abundances of *Fragilaria* and *Ulnaria*, so that they may be generalized at least to all freshwater diatoms, and likely to the majority of non-model unicellular life.

In conclusion, we showed that V3-V4 or V4 18S rRNA amplicons can be reliably used for in situ distinguishing between closely related diatoms, although not quantitatively. Qualitative comparison shows that Baikalian populations of *Ulnaria acus* and *Fragilaria radians* differ in their distribution in both space and time.

**Supplementary Materials:** The following supporting information can be downloaded at: https://www.mdpi.com/article/10.3390/d15020280/s1: Table S1: Sequences and abundances of *Fragilaria/Ulnaria* ASVs produced from samples taken near the settlement of Bolshiye Koty, March 2020, and in an artificial mock community, by various methods and amplicons. Table S2: Taxonomy and distribution of 18S ASVs in samples taken near the settlement of Bolshiye Koty, March 2020, and in an artificial mock community.

**Author Contributions:** Conceptualization, A.M. (Alexey Morozov); methodology, A.M. (Alexey Morozov), A.M. (Artem Marchenkov), Y.G.; software, A.M. (Alexey Morozov); formal analysis, A.M. (Alexey Morozov); resources, Y.Z.; writing—original draft preparation, A.M. (Alexey Morozov) and A.M. (Artem Marchenkov); writing—review and editing, A.M. (Alexey Morozov), A.M. (Artem Marchenkov), Y.G., Y.Z., D.P.; visualization, A.M. (Alexey Morozov); supervision, Y.G. and D.P. All authors have read and agreed to the published version of the manuscript.

**Funding:** The study was funded by the Ministry of Science and Higher Education of the Russian Federation, project number 0279-2021-0009.

**Institutional Review Board Statement:** Not applicable.

**Data Availability Statement:** The sequencing data for strain mixture and Bolshiye Koty samples are available at NCBI SRA under project ID PRJNA666300.

**Acknowledgments:** The authors are grateful to Natalia Annenkova and Ivan Mikhailov for providing their metadata and for help with the technical details of their studies. The authors would like to thank Irkutsk Supercomputer Center of SB RAS for providing access to HPC-cluster "Akademik V.M. Matrosov".

**Conflicts of Interest:** The authors declare no conflict of interest. The funders had no role in the design of the study; in the collection, analyses or interpretation of data; in the writing of the manuscript; nor in the decision to publish the results.

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
