# Peer review of "Revealing the Differences in Ulnaria acus and Fragilaria radians Distribution in Lake Baikal via Analysis of Existing Metabarcoding Data"

_diversity, doi:10.3390/d15020280_

Round 1
Reviewer 1 Report
This study is interesting and well written. Although the topic and the conclusions drawn from the research are expected and already discussed by other papers, the authors of this study managed to give new insights and focusing on specific Baikal diatoms offer region specific value. Since the study strongly depends on methodology, I think that part of the study needs improvement in giving more details and better explaining data processing steps.

Author Response
We are grateful for the review. As for the methods, the data analysis section has been expanded and should now be sufficient for reproducing our work. Also, please note that pipelines described in section 2.1 (for test data) and in section 2.2 (for existing metabarcoding datasets) were exactly identical, and the mention of this fact was added to section 2.2.
Reviewer 2 Report
It is difficult to judge this manuscript, because one of the central assumptions—that Ulnaria acus and Fragilaria radians are two distinct species rather than a reflection of intraspecific variation—is not verifiable to the reader. My gut instinct is to reject this manuscript, pending the release of the cited manuscript. That being said, there does appear to be signal in the data suggesting some ecological and/or geographical sorting…but the inferences that can be drawn from these data are limited without more information about the taxa in question. There is value in this manuscript, but it would be much more valuable once all the information about this study system is made available to the public. With all that in mind, I’m afraid I have to recommend this manuscript be rejected, but resubmitted once the comparative study between Ulnaria acus and Fragilaria radians is published.
Introduction
Lines 24-26: What is the source of the “Synedra acus subsp. radians Skabitsch” name? Williams and Round (1988) used Synedra radians Kutzing as the basionym for Fragilaria radians…and neither name is included in the transfers proposed in the cited Bukhtiyrova and Compere paper.
Lines 41-45: This is problematic. The authors are telling us that Fragilaria radians and Ulnaria acus are distinct but nearly indistinguishable entities, but their evidence for this is unpublished. How can the reader judge this statement, or evaluate the utility of this manuscript, if the central tenant of the study is based on an unsupported inference? I’m not suggesting that the authors are incorrect in their inference that the aforementioned taxa exist as two distinct entities, but the authors provide the reader with no pathway to verify this statement.
Line 48: “…based on the assumption of a single species, using…wider Synedra sp./Fragilaria sp. OTUs that include both F. radians and U. acus…” This needs a bit more explanation. Are the authors suggesting here that previous environmental DNA and barcoding studies treated Fragilaria radians and Ulnaria acus as a single taxonomic entity because the barcode sequence for both taxa was included among specific OTUs identified in the cited studies? Or because the barcode sequences for F. radians and U. acus were among the OTUs of Fragilaria and Synedra which were not identified to species?
Materials and Methods
Line 96: “Three hundred and twelve bp…” or “A 312 bp rbcL fragment…” Should not start a sentence with a numeral.
Line 121: So, just to be clear—OTUs and ASVs were assigned to genera based on a 97% sequence similarity? Is this an a priori assumption based on a previous study? Presumably assignment to species was at 100% only?
Line 129: Same question as above, regarding assignment of taxonomy to OTUs and ASVs—98% sequence similarity was the cut-off to genus?
Results
Line 158: Which two Ulnaria clades? Three Ulnaria taxa are mentioned.
Line 163: “…both markers…” Three markers were sequenced. Does this refer only to the 18S markers mentioned in the previous sentence or the V3-V4 and rbcL illustrated in Figure 1?
Line 164: Define “valid”. What, specifically, was being tested? What result defines validity for studying the relative abundance of U. acus and F. radians?
Line 191: The Introduction stated that the taxon would be referred to as “Fragilaria radians”.
Discussion
Line 234: “…ecological differences within genera are usually not very important…” Please explain this. Are the authors actually suggesting that ecological niche differentiation plays no role in speciation, or is this a convention of metabarcoding studies due to the paucity of verified reference sequence data relative to diatom diversity?
Lines 252-260: Isn’t this (resolving taxa in microscopy vs. barcode sequences) the same problem: the lack of a common reference and toolkit? Taxa can only be consistently resolved under the microscope if the same reference images and tools (100x oil objective lenses, DIC filters, electron microscopy), but they also can only be resolved by sequence data with a publicly-available, vouchered sequence library and an appropriate marker.
Line 336: To be fair, dozens of studies exist in the diatom literature which show “closely related diatoms” (a relative term which is difficult to compare across studies…) can be distinguished by V3-V4 18S and rbcL markers. More compelling is the evidence that spatially and/or ecologically distinct diatoms in the genus Ulnaria or Fragilaria (species or populations, depending on the sequence annotation) seem to exist in Lake Baikal.
Round 2
Reviewer 2 Report
I appreciate the authors' efforts in responding to my concerns and suggestions. Though I am still hesitant to recommend this manuscript for publication without seeing the sister manuscript documenting the distinction between Ularia acus and Fragilaria radians, I cannot in good conscience reject this particular manuscript, which makes a decent case for en ecology-based changeover in the Baikal flora...whether that is species or population-based remains to be seen.